# The condition for dynamic stability in humans walking with feedback control

Hendrik Reimann[1]*, Sjoerd M. Bruijn[2,3]

**1** Department of Kinesiology and Applied Physiology, University of Delaware, Newark, Delaware, United States of America, **2** Department of Human Movement Sciences, Faculty of Behavioural and Movement Sciences, Vrije Universiteit Amsterdam, Amsterdam, The Netherlands, **3** Institute of Brain and Behavior, Amsterdam, The Netherlands

* hendrikreimann@gmail.com

**Data Availability Statement:** All code for model simulation and figure generation is available at github.com/hendrikreimann/WalkingModelPaper2023.

## Abstract

The walking human body is mechanically unstable. Loss of stability and falling is more likely in certain groups of people, such as older adults or people with neuromotor impairments, as well as in certain situations, such as when experiencing conflicting or distracting sensory inputs. Stability during walking is often characterized biomechanically, by measures based on body dynamics and the base of support. Neural control of upright stability, on the other hand, does not factor into commonly used stability measures. Here we analyze stability of human walking accounting for both biomechanics and neural control, using a modeling approach. We define a walking system as a combination of biomechanics, using the well known inverted pendulum model, and neural control, using a proportional-derivative controller for foot placement based on the state of the center of mass at midstance. We analyze this system formally and show that for any choice of system parameters there is always one periodic orbit. We then determine when this periodic orbit is stable, i.e. how the neural control gain values have to be chosen for stable walking. Following the formal analysis, we use this model to make predictions about neural control gains and compare these predictions with the literature and existing experimental data. The model predicts that control gains should increase with decreasing cadence. This finding appears in agreement with literature showing stronger effects of visual or vestibular manipulations at different walking speeds.

## Author summary

The walking human body is mechanically unstable and humans frequently lose upright stability and fall while walking. Stability of human walking is usually analyzed from a biomechanical perspective. We argue that sensorimotor control is an essential aspect of walking stability. We model a walking system as a combination of inverted pendulum biomechanics and a neural feedback controller for foot placement and analyze the properties of this hybrid dynamical system. We find that there is always a periodic orbit and derive a criterion for the asymptotic stability of this periodic orbit. This analytic criterion allows us to characterize the region in the parameter space where the walking system is stable. We use these theoretical results to analyze stability of human walking depending

**Funding:** The author(s) received no specific funding for this work.

**Competing interests:** The authors have declared that no competing interests exist.

on different sensorimotor control gains. The model predicts that control gains should be larger for slower-paced walking, which is partially consistent with the available experimental data.

## 1 Introduction

The upright human body is unstable, and humans sometimes fall. People tend to fall more often as they get older [1, 2], and those with certain neuromotor impairments, such as cerebral palsy or Parkinson's disease, tend to fall more often [3–6]. Staying upright seems harder in some situations, such as when the eyes are closed, when standing on foam or when adopting a tandem stance [7]. To quantify these effects, we need a measure for *stability*.

The most widely used measure of stability is the *Margin of Stability* (MoS), introduced by [8]. This measure combines the position and velocity of the body center of mass (CoM) to a measure called the *extrapolated center of mass* (XCoM) and analyzes it with respect to the base of support. If the XCoM is within the base of support, it is possible to maintain upright stability by moving the center of pressure (CoP) within the base of support. If the XCoM is outside the base of support, the accelerations generated by CoP movements are too restricted and will not be sufficient to maintain upright stability without adjusting the base of support by taking a step, or generating angular momentum around the CoM. The MoS is the signed distance from the XCoM to the limit of the base of support. It is defined as positive when the XCoM is within the base of support and negative when it is outside. The MoS is proportional to the impulse required to destabilize the upright body and force it to take a step or fall. If that impulse is higher rather than lower, then the body requires a larger perturbation to make it fall, and we call it "more stable". In that sense, the MoS measures the degree of mechanical stability for the standing human body.

Maintaining upright stability requires continuous control by the nervous system. A positive MoS indicates that it is possible to remain upright in the near future without having to take a step, by modulating the CoP within the base of support. The MoS measure is agnostic to the specific details of *how* the nervous system might achieve that. The control problem for the nervous system is to estimate the current state of the body relative to the base of support, based on sensory data, and transform that into descending motor commands that activate muscles to generate forces against the ground and shift the CoP in a way to prevent falling. As stated above, the MoS is a purely biomechanical measure. It ignores the control problem of *how* the CoP is modulated and only states that *it is biomechanically possible* to remain upright without taking a step by moving the CoP within the base of support.

In standing, ignoring the control aspect is reasonable, because the biomechanical stability condition of "can be stabilized without taking a step" is a meaningful criterion that is relevant to the studied behavior, without having to specify explicitly *how* this control system that achieves stabilization in the given situation is structured. But in walking, taking steps is a regular and essential part of the movement. Still, the MoS has been widely applied to quantify stability in walking [9]. Using the MoS in walking has had some results that are incongruent with its definition. E.g. people tend to walk with larger MoS in the presence of perturbations, and populations with stability problems such as Parkinson's Disease or cerebral palsy also tend to walk with larger average MoS [10]. These effects are at odds with the concept of the MoS as a measure of stability, because we would expect any such measure to show a decrease, rather than increase in stability, when adding perturbations or looking at populations with increased fall risk.

We postulate that the main reason that the MoS does not work well as a stability measure in walking is that it neglects the control aspect. This is reasonable in standing, because "can the body be stabilized without taking a step?" is a meaningful question to ask. In walking, the qualifier *without taking a step* is not useful, because the answer is "no" most of the time, and even when it is "yes", we would take a step anyways, because the goal is locomotion. When the system is free to take a step to any location at any point in time, on the other hand, the body can always be stabilized, e.g. by taking a step instantaneously and placing the CoP exactly on the XCoM [11, 12]. For the stability question to be meaningful, we need to consider limitations about *when* and *where* a step can be taken. In other words, we need to account for how foot placement is controlled.

In this paper, we attempt to take a first step towards answering the question "how can the body be stabilized" for walking. Our approach to this question is novel in two critical aspects. First, we account for neural control by postulating a control law for foot placement that determines when and where to a step is taken. For the choice of control law, we will follow experimental results showing that foot placement location is largely determined by the state of the center of mass at midstance [13], and model this behavior with a proportional-derivative controller. Second, instead of focusing on a body state at a given instance in time, we will analyze the walking system as a whole. We do this because loss of stability in walking is sometimes not instantaneous, but happens gradually across multiple steps. Thus, the research question we ask here is "How does the neural controller of a walking system need to be structured for the gait to be stable".

In the following sections, we will first define the walking system we are analyzing. Biomechanically, we use the same approach as the MoS and approximate the body as a single point mass with linearized equations of motion. The control is represented by a proportional-derivative feedback law for foot placement location based on the position and velocity of the center of mass at midstance. Next, we will define what we mean for such a walking system to be "stable" in the context of this article. This formal definition will allow us to provide conditions that the system parameters have to meet for the system to be stable. We present the results as formal statements and provide the derivations in S1 Text.

Along the formal results, we provide examples with the goal of making the technical results accessible to a broader audience. Finally, we show how the theoretical model relates to actual walking by making predictions for how control parameters for stability in humans should change at different cadences and comparing them to experimental data.

## 2 Methods

### 2.1 The walking system

In bipedal locomotion, during the time intervals when a single leg is in contact with the ground, the movement of the body is largely ballistic, with gravity being the only major force acting. Here, we model this movement using several common simplifications. We assume that the body is a point mass that is supported by rigid legs, commonly called a *single-link inverted pendulum*. At any time, exactly one leg has contact with the ground, and the location of that contact point is changed when taking a step. The legs are massless and moving the contact point to a new location is instantaneous. Steps are taken after a constant time interval and the location of the new contact point at each step is determined by a control law. In humans, this control law would be implemented at different levels of the nervous system, and could even arise in part from passive dynamics [14]. Fig 1 shows a schematic overview of the model.

We will formalize both the biomechanics and the control below. Locomotion is mainly a two-dimensional motion, and we will deal with the vertical dimension only implicitly here. In

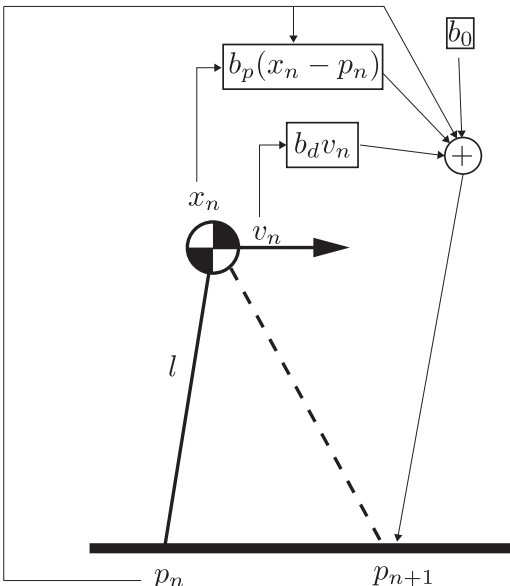

**Fig 1. Schematic overview of the model.** The biomechanical state of the system during the $n$-th step is modeled by position, $x_n$, and velocity, $v_n$ of a point mass connected to the ground via a rigid, massless leg of length $l$ at point $p_n$. The controller selects a foot placement location based on a proportional-derivative controller with the difference of the point mass from the contact point, $x_n - p_n$ and its rate of change, $v_n$, with gain factors $b_p$ and $b_d$, as well as a constant offset, $b_o$.

typical walking, movement in the horizontal plane can be separated into a main direction of forward progression and a second direction in which the body oscillates sideways with little net movement over time. We will refer to these directions as the anterior-posterior and the medial-lateral directions, although these terms refer to properties of the body that the point-mass model does not have. The dynamics of the single-link inverted pendulum are identical in these two directions, and not coupled, but differences in control can lead to the different large-scale movement patterns of forward progression and sideways oscillation described above. In the anterior-posterior direction, the body progresses consistently forward, with the body mass reaching and moving over the foot contact point at each step. In the medial-lateral direction, the body alternates between two directions of movement, with the body mass approaching the foot contact point, then coming to a halt and changing movement direction before each new step. We refer to these two movement patterns as *progressive* and *alternating* stepping.

**Biomechanics.** The dynamics of the single-link inverted pendulum have been analyzed in detail elsewhere [8, 15, 16], and here we use the equations derived in this analysis. Briefly, the body is approximated by a single point-mass on a rigid, mass-less leg. The state of this system is described by the position, $x$, and velocity, $v$, of the body center of mass, as well as the location of the contact point between the leg and the ground, $p$. The location of the contact point changes instantaneously when taking a step but is constant between steps. The state of the point-mass changes continuously over time. We linearize the equations of motion [15], which effectively movement in the vertical direction and constrains the CoM to move in a horizontal plane parallel to the ground. This linearized system, which we will simply refer to as *inverted pendulum* from now on, is determined by the equations of motion

$$\dot{x} = v, \qquad \dot{v} = \omega^2(x - p) \tag{1}$$

where $\omega = \sqrt{\frac{g}{l}}$ is the eigenfrequency of the inverted pendulum, $l$ is the length of the rigid leg and $g = 9.81 \text{ m s}^{-2}$ the acceleration from gravity.

For any initial conditions $x_0$, $v_0$ and contact point location $p$, solutions to this system are given by the hyperbolic functions

$$x(t) = p + (x_0 - p)\cosh(\omega t) + \frac{v_0}{\omega}\sinh(\omega t)$$
$$v(t) = (x_0 - p)\sinh(\omega t)\omega + v_0\cosh(\omega t)$$

(2)

(see [8, 17] and Section 1) in the S1 Text.

**Control.** The inverted pendulum is controlled by taking steps. Steps are instantaneous shifts of the contact point, $p$, to a new location. We note here that steps in our model do not cause a collision, and hence no loss in energy, following similar models [8, 15, 16]. Here, we assume that the duration of each step, $T_{\text{step}}$, is constant and the only degree of freedom for the controller is choosing the location of the step. Foot placement is critical for control of stability [18]. Based on [13], we postulate that a new foot placement location is chosen using a proportional-derivative control law, which uses as input the CoM position and velocity relative to the CoP at midstance of the current step. To formalize this, we introduce new variables for the position, $x_n$, and velocity, $v_n$, of the CoM at half-time of the $n$-th step, as well as for the position of the contact point, $p_n$, during the $n$-th step. By "half-time" we refer to the point in time in the middle between two steps, i.e. at time $t = \frac{1}{2}T_{\text{step}}$ after a step. We will also refer to this point as *midstance*, but note that it is defined differently in the model from how midstance is usually determined in experimental data [13].

Based on the considerations above, we define the neural control law for foot placement based on the state of the CoM at midstance in *progressive* walking in the anterior-posterior direction as

$$p_{n+1}^{\text{ap}} = p_n + b_o + b_p(x_n - p_n) + b_d v_n,$$

(3)

where $b_p$ is the proportional and $b_d$ the derivative control gain. The additional parameter $b_o$ is a constant offset to foot placement location. It can be used to modulate average step length or width, but, as we will show later, has no bearing on the stability of the system.

For *alternating* walking in the medial-lateral direction, we add a factor that alternates the direction of the offset, changing the equation to

$$p_{n+1}^{\text{ml}} = p_n + (-1)^n b_o + b_p(x_n - p_n) + b_d v_n.$$

(4)

The only difference between the two versions of the system describing *progressive* walking in the anterior-posterior direction and *alternating* walking in the medial-lateral direction is the addition of the alternating sign of the offset. As we will see later, this has no effect on many system properties, including stability. In the interest of keeping notation simple and to avoid excessive subscripts in the analysis, we will generally only specify the direction of movement in cases where it is relevant. We also note that analyzing movement in the anterior-posterior and medial-lateral directions as separate, one-dimensional dynamical systems is possible due to the simplifying assumptions made here, specifically the linearization, lack of swing leg dynamics and constant step time. The actual human walking through physical space is, of course, three dimensional, and our dynamical system define by Eq 1 describes one of the three dimensions of physical space.

**Walking solutions.** The system described by Eqs 1–4 is a single-link inverted pendulum controlled by a linear proportional-derivative controller which selects foot placements after a

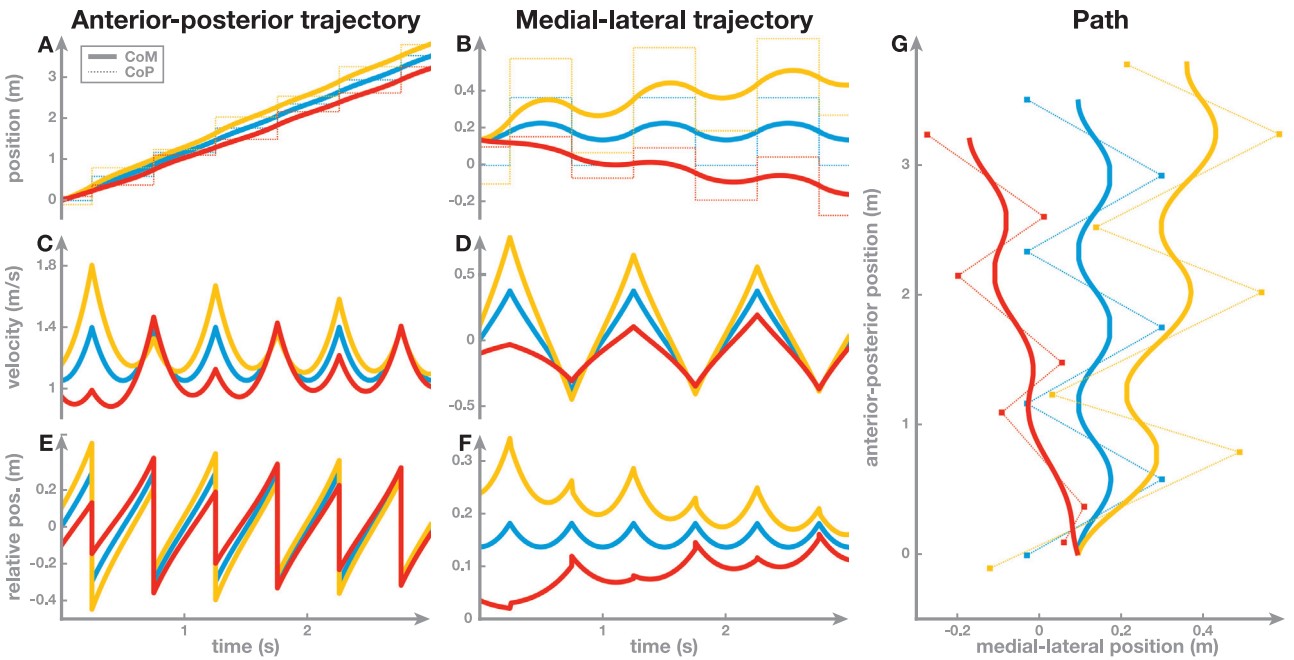

**Fig 2. Example solutions for the walking system with three different initial conditions.** The left column shows progressive walking in the anterior-posterior direction, the center column alternating walking in the medial-lateral direction, and the right panel combines both directions to show a top-down view of the path. Panels A and B show the absolute position of the CoM, $x(t)$, (solid) and CoP, $p(t)$, (dotted) vs. time, Panels C and D show the CoM velocity, $v$, and Panels E and F show the position of the CoM relative to the CoP, $q(t) = x(t) - p(t)$. The initial conditions were chosen slightly different, but the system parameters were the same in all three cases.

constant step time. This is a hybrid dynamical system with two continuous state variables, $x$ and $v$, and one discrete state variable, $p$, that is piecewise constant in the time intervals of length $T_{step}$ between two steps. In the following, we will refer to this as a *walking system*. A walking system is parameterized by a single biomechanical parameter, the eigenfrequency, $\omega = \sqrt{gl^{-1}}$, of the inverted pendulum, where $g$ is gravitational acceleration and $l$ the effective leg length, i.e. the vertical distance from the CoM to the contact point, as well as four control parameters, $b_o$, $b_p$, $b_d$ and $T_{step}$. Such a walking system models walking *in a single direction only*, but two systems can be combined to describe walking across a plane spanned by the usual anterior-posterior and medial-lateral directions.

Fig 2 shows example solutions of such a walking system, for the parameters $\omega = 3.13$ s$^{-1}$, $b_o = -0.05$ m (progressive), $b_o = 0.02$ m (alternating), $b_p = 2.5$, $b_d = 0.6$ m and $T_{step} = 0.5$ s. The initial time is set to the midstance of a step, so that the first step occurs at $t = \frac{1}{2}T_{step} = 0.25$ s. All solutions use the same values for biomechanical and control parameters, but different initial conditions. The left column shows solutions for a system using progressive stepping in the anterior-posterior direction, the center column shows solutions for a system using alternating stepping in the medial-lateral direction. In both cases, the top graph shows position, the middle graph shows velocity and the bottom graph shows the position of the body mass relative to the contact point, $x - p_n$. The right panel combines both directions to show a top-down view of the horizontal plane. Steps are taken after $T_{step} = 0.5$ s, indicated by the instantaneous changes of the CoP location. In the top-down view on the right panel, the discrete CoP locations are shown as solid squares, with the dotted lines indicating the instantaneous switches between subsequent foot placements.

These examples illustrate that a periodic solution (i.e. periodic orbit) exists for the chosen parameter values. This periodic orbit is the blue curve at the center of each graph. In the medial-lateral direction, both position and velocity of this solution are periodic with a period of $2T_{\text{step}} = 1$ s. In the anterior-posterior direction, the position, $x(t)$ is not periodic, but consistently increases, as expected from a system walking forward. However, the position relative to the contact point, $x(t) - p_n(t)$, is periodic, as well as the velocity, with a period of $T_{\text{step}} = 0.5$ s. We will analyze periodic orbits more formally below.

For the example shown in blue in Fig 2, we chose the initial conditions exactly right so that the solution would be periodic (see Result 1 below for how to determine the periodic solution to find such an initial condition). For the other two example solutions, shown in red and yellow, we slightly changed the initial conditions, adding $\Delta p = \pm 0.1$ m to the initial position and $\Delta v = \pm 0.1$ m s$^{-1}$ to the initial velocity to showcase how this changes the solutions over time. These different initial conditions lead to movements that are markedly different over the first few steps. The differences in velocity mostly disappear towards the end, as the initially perturbed yellow and red solutions become more similar to the blue solution (Fig 2C and 2D). For position, substantial differences in the absolute position remain (Fig 2A and 2B), but the position of the CoM *relative* to the CoP becomes similar to the periodic orbit again (Fig 2E and 2F). This shows that although the red and orange solutions are different, they relax to the same periodic orbit after some time, in relative terms. The path in the horizontal plane formed by the combination progressive stepping in the anterior-posterior direction and alternating stepping in the medial-lateral direction is shown in Fig 2G.

The walking solutions depend not only on the initial conditions, but also on the values of the control parameters $b_o$, $b_p$, $b_d$ and $T_{\text{step}}$. This blue solution in this example corresponds to a periodic orbit in the phase space of the system, and this periodic orbit is asymptotically stable, indicated by the fact that the system returns to the periodic orbit after small changes in the initial conditions (red and yellow lines). In the next section we will show that for all possible values of the control parameters a periodic orbit exists. In the subsequent section we will analyze the asymptotic stability of these periodic orbits depending on the values of the control parameters.

## 2.2 Periodic orbits

In the example above, we chose one specific set of values for the parameters, $\omega$, $b_o$, $b_p$, $b_d$ and $T_{\text{step}}$, and simulated solutions for different initial conditions. We found that one of these solutions was periodic and the other solutions, starting from slightly different initial conditions, relaxed towards this periodic orbit. Now, we will analyze the system for *any* value for the control parameters, with the goal of determining (1) whether periodic orbit exist for the given system and (2) whether any periodic orbit is asymptotically stable. We will find that any walking system has a periodic orbit. We will give formulas for the state of the system at midstance of the step when it is on this periodic orbit, which we used to generate the initial condition for the example above (blue lines in Fig 2. Finally, we will find that only some of these periodic orbits are asymptotically stable and determine the region of the parameter space in which this is the case. We will develop the concepts and then state these results formally, while providing the technical derivations in the S1 Text.

For any initial condition, the solutions of the system are given by Eq 2. Without loss of generality, we will analyze solutions of the system starting at midstance, so that the first step happens at time $t_s = \frac{1}{2}T_{\text{step}}$ and the next midstance is reached at time $t_m = T_{\text{step}}$. To simplify

notation, we choose short-hand variables for the hyperbolic functions after half a step as

$$c = \cosh\left(\omega \frac{1}{2} T_{\text{step}}\right), \qquad s = \sinh\left(\omega \frac{1}{2} T_{\text{step}}\right). \tag{5}$$

Since we are interested in the position of the CoM relative to the contact point, rather than the absolute CoM position, we define a new variable,

$$q(t) = x(t) - p(t) \tag{6}$$

for the relative position.

**Result 1** *Consider any walking system, given by a set of parameters $\omega$, $b_o$, $b_p$, $b_d$ and $T_{\text{step}}$. This system has a periodic orbit for $(q, v)$. For systems with progressive step control, the period of this orbit is $T_{\text{step}}$ and the state at midstance is given by*

$$q^{\text{ref,ap}} = 0, \qquad v^{\text{ref,ap}} = \frac{b_o}{2\frac{s}{\omega} - b_d}. \tag{7}$$

*For systems with alternating step control, the period of the orbit is $2T_{\text{step}}$ and the state at midstance is given by*

$$q_n^{\text{ref,ml}} = (-1)^n \frac{b_o}{2c - b_p}, \qquad v^{\text{ref,ml}} = 0. \tag{8}$$

We use the superscript "ref" for *reference* here, since going forward we will use the state at midstance of the periodic orbit as a reference point to analyze other solutions at midstance. This result shows that at midstance of the periodic orbit, the CoM of the walking system will be exactly above the contact point in the anterior-posterior direction ($q^{\text{ref,ap}} = 0$). In the medial-lateral direction, the walking system will reach zero velocity at midstance of the periodic orbit ($v^{\text{ref,ml}} = 0$). The non-zero values in each case change depending on the system parameters according to Eqs 7 and 8.

Fig 3 shows an example of the periodic solution in phase space for progressive walking (Fig 3A) and alternating walking (Fig 3B). The control parameters in this example are the same as in the previous example, with solutions shown in Fig 2. The difference here is that instead of showing the solutions against time or the paths in space, we show the orbits in phase space, i.e. CoM position, $x$ on the horizontal and velocity, $v$, on the vertical axis. The blue line shows how the state of the system develops in phase space for the first second ($2T_{\text{step}}$ in this example). In both panels, the initial condition is on the very left of the blue curve, and the gray arrows inside the blue curve indicate the direction of movement along the orbit, as well as the time, with arrows placed evenly at 0.05 s intervals. In progressive walking, the position continuously increases, with velocity going up and down but consistently positive (Fig 3A). In alternating walking, both position and velocity go up and down, forming a closed loop, with the state returning to the initial condition at time $2T_{\text{step}}$. The contact point at each step, $p_n$, are shown as orange dots on the horizontal. At time $t = \frac{1}{2} T_{\text{step}}$, the contact point is changed from $p_0$ to $p_1$, according to Eqs 3 (progressive stepping, Fig 3A) and 4 (alternating stepping, Fig 3B).

The dynamics of the overall system are illustrated by the thin gray flow lines. These flow lines are examples of other orbits, with dots at 50 ms intervals indicating timing. With the change in contact point, the dynamics also change instantaneously, according to Eq 1. This change in dynamics is indicated by the vertical gray lines. Note that this limit line, where the illustration changes from showing the flow lines of the system for $p_0$ to the system for $p_1$, is only a choice of the illustration. The change of dynamics is a result of the change in foot placement, which happens at a specific point in time, rather than when the CoM position, $x$, reaches

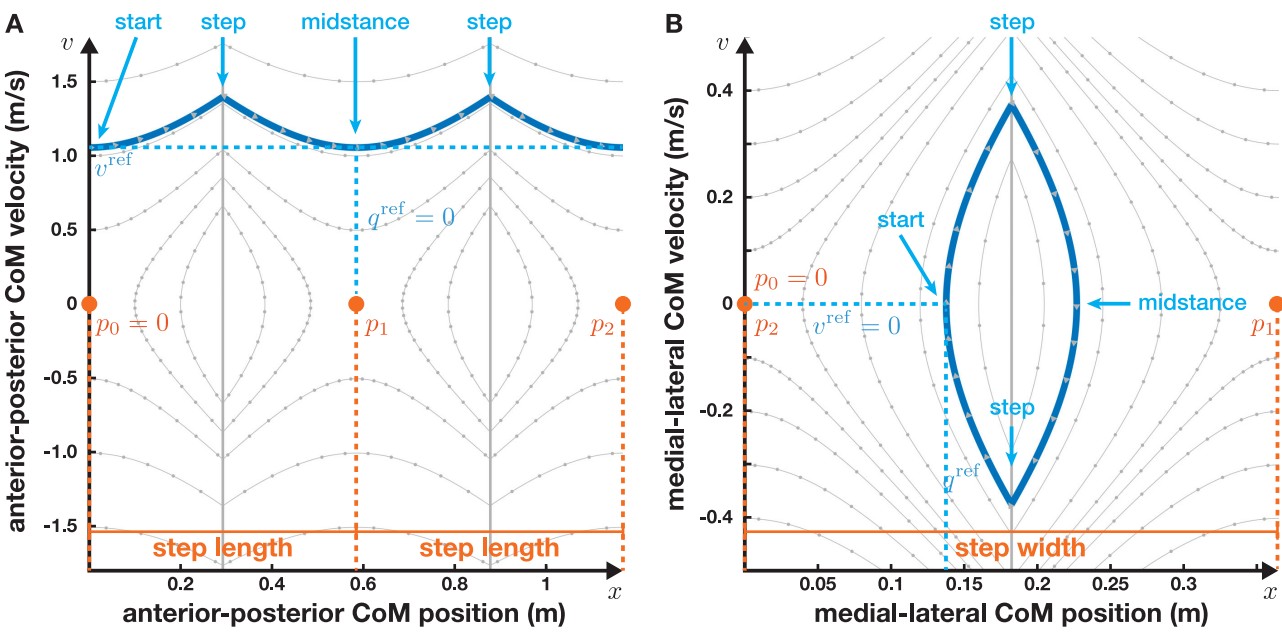

**Fig 3. Phase plot of the periodic orbits for progressive (Panel A) and alternating stepping (Panel B).** The horizontal axis is position of the CoM, the vertical axis is velocity. The thick blue line is the orbit. Here, the solutions are shown over 1 s, starting at midstance and including two steps. Gray arrows in the blue orbit indicate direction and time, spaced 0.05 s apart. The thin gray curves are other orbits. The vertical gray lines indicate an instantaneous change in the dynamics from taking a step, and the orbit has a cusp at these points. The contact point, $p_n$, is shown in orange for each step.

the vertical line. At this point in time, the whole system dynamics changes, including the parts beyond the area shown in Fig 3. For the orbit shown in blue, the combination of timing and new foot placement location line up so that the velocity at the end of step is exactly the same as at the beginning of the step, which makes this the periodic orbit.

While some of the other flow lines shown in gray *look* like they might be periodic, they are not. With the chosen control parameters, the orbit lines up to be periodical only if the state is on the specific point shown here when it is time for a step. If it is at any other point in phase space, including any other point on the gray vertical line, the new foot placement location, determined by Eqs 3 and 4 would be different in a way that makes the solution as a whole non-periodic.

## 2.3 Stability

We analyze the local stability of the walking system, i.e. how it responds to small perturbations. Note that this is conceptually different from related concepts such as global stability or viability [19]. We have shown that each walking system, defined by a set of parameters $\omega$, $b_o$, $b_p$, $b_d$ and $T_{\text{step}}$, has a periodic orbit. For the parameter set that we have chosen in the examples above, shown in Figs 2 and 3, this periodic orbit is asymptotically stable, in the sense that when the initial condition is not on the periodic orbit, the system will relax towards the periodic orbit over time. This is not necessarily the case for other choices of parameters.

**Examples of stable and unstable walking.** Fig 4 shows example solutions for four walking systems taking alternating steps in the medial-lateral direction. The system parameters are the same in all four examples, except for the derivative control gain, $b_d$, which is set to four different values of $b_d$ = 0.3, 0.6, 1.1 and 1.5 s. This parameter determines how strongly the controller responds to a change in the CoM velocity at midstance (see Eqs 3 and 4). As implied by Result 1, each walking system has a periodic orbit. This orbit is the same for all four systems in this

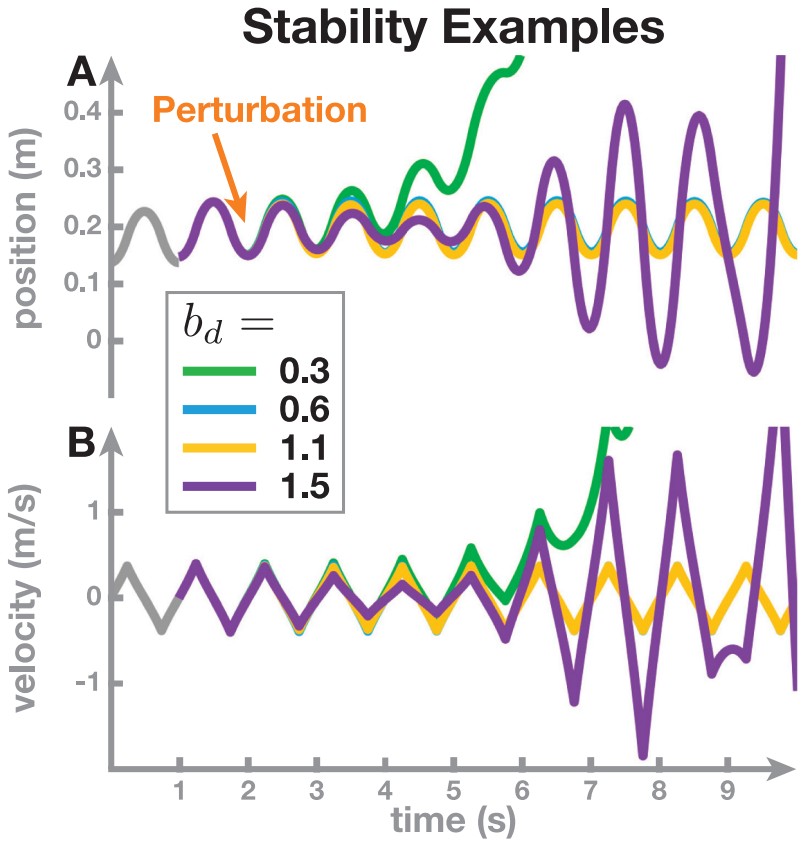

**Fig 4. Examples of solutions with four different values of the derivative gain parameter, $b_d$.** Panel A shows the CoM position, Panel B the CoM velocity, vs. time. The initial condition is the same in all four systems, and the solution over the first second is also identical (gray). At $t = 1$ s, a small perturbation is applied, which changes the solution of the system depending on the value of the control gain, $b_d$.

example, because the reference configuration that determines the periodic orbit does not depend on the parameter $b_d$ for alternating stepping, as seen from Eq 8. Here we simulated all four systems with this shared reference state as initial condition, and they follow the periodic orbit for the first second of the movement, taking two steps (gray lines in Fig 4). At time $t = 1$ s, we simulated a small perturbation by increasing the CoM position by 1 cm. The subsequent nine seconds show that the four systems react to this perturbation very differently. The first system, with $b_d = 0.6$, is the one we already analyzed in the previous section. The solution of this system (blue) changes very little in response to the perturbation at 1 s and relaxes back to the periodic orbit quickly afterwards. A small increase in derivative gain to $b_d = 1.1$ in the second system (yellow) changes very little, with the curves for position and velocity virtually identical. Increasing the derivative control gain further to $b_d = 1.5$, however, leads to a substantially different effect in the third system (purple). The system oscillates with increasing amplitude, and does not relax back to the periodic orbit. The control gain is too large, so the system over-corrects on each step, responding too strongly to the velocity difference at midstance. In the fourth system (green), we changed the control gain in the opposite direction, to $b_d = 0.3$. This also leads to failure. The state starts to diverge soon after the perturbation, becoming more and more positive. The control gain is too small, so the system under-corrects on each step and the velocity keeps building up from step to step.

**When is a walking system stable?.** While all walking systems have a periodic orbit, these examples show that this periodic orbit is not necessarily stable. Some periodic orbits are "stable" only in the sense of a pencil standing on its tip: while it is theoretically possible, any minuscule deviation from the upright will lead to a fall. In this section, we analyze the stability of a walking system, in the sense of whether they will return to a periodic orbit after a perturbation. Our goal is to characterize how this stability depends on the system parameters. We start by introducing a measure for the deviation from the periodic orbit of the system at midstance of the $n$-th step

$$\delta_n = \begin{pmatrix} q_n - q^{\text{ref}} \\ v_n - v^{\text{ref}} \end{pmatrix}, \tag{9}$$

where we omit the index, $n$, in $q_n^{\text{ref,ml}}$ in alternating systems for brevity. We call a walking system *stable* if and only if for small perturbations of the system at initial midstance, $\delta_0$, the deviation at subsequent steps, $\delta_n$, goes to zero, i.e.

$$\lim_{n \to \infty} \delta_n = 0. \tag{10}$$

Note that this definition of stability is the same as the *asymptotic stability* of the periodic orbit. We will sometimes add the qualifier "asymptotically" to make this clear.

To analyze the stability of a system, we need to calculate how the deviation at midstance of one step determines the deviation at the midstance of the next step. In other words, the deviation is an error, and we need to calculate how that error propagates from one midstance to the next. We do this in the following Result, which also draws conclusions from the error propagation about stability of the system.

**Result 2** *For any walking system given by a set of control parameters $\omega$, $b_o$, $b_p$, $b_d$ and $T_{\text{step}}$, the error propagation from the n-th to the n + 1-th step is given by the linear mapping*

$$\delta_n \mapsto \delta_{n+1} = A\delta_0, \tag{11}$$

*with*

$$A = \begin{pmatrix} c^2 + s^2 - cb_p & \left(2\frac{s}{\omega} - b_d\right)c \\ (2c - b_p)s\omega & c^2 + s^2 - s\omega b_d \end{pmatrix}. \tag{12}$$

*The walking system is asymptotically stable if and only if the largest absolute eigenvalue of A is less than 1.*

**How do the control parameters affect stability?.** We have now established a criterion for the stability of a walking system. We have already seen in the example above in Fig 4 that when we increase the derivative gain, $b_d$, and hold all other parameters constant, the walking system changes from unstable to stable, and back to unstable. Result 2 allows us to determine the stability of a walking system directly from the system parameters, without having to calculate different solutions, by calculating the eigenvalues of $A$. Note that stability depends on the transition matrix, $A$, which in turn depends on the body eigenfrequency, $\omega$, the control gains, $b_p$ and $b_d$, and the step time, $T_{\text{step}}$, via $c$ and $s$ from Eq 5. The constant offset parameter, $b_o$, on the other hand does *not* affect the stability of the system. Since $\omega$ depends only on the biomechanics of the body, we will assume that it is constant. For now we will also keep the step time, $T_{\text{step}}$, constant. The following result shows how stability depends on the remaining parameters, the control gains $b_p$ and $b_d$. Note that while this result can tell us whether the walking system is asymptotically stable or not, this does not in any way relate to how large of a perturbation the

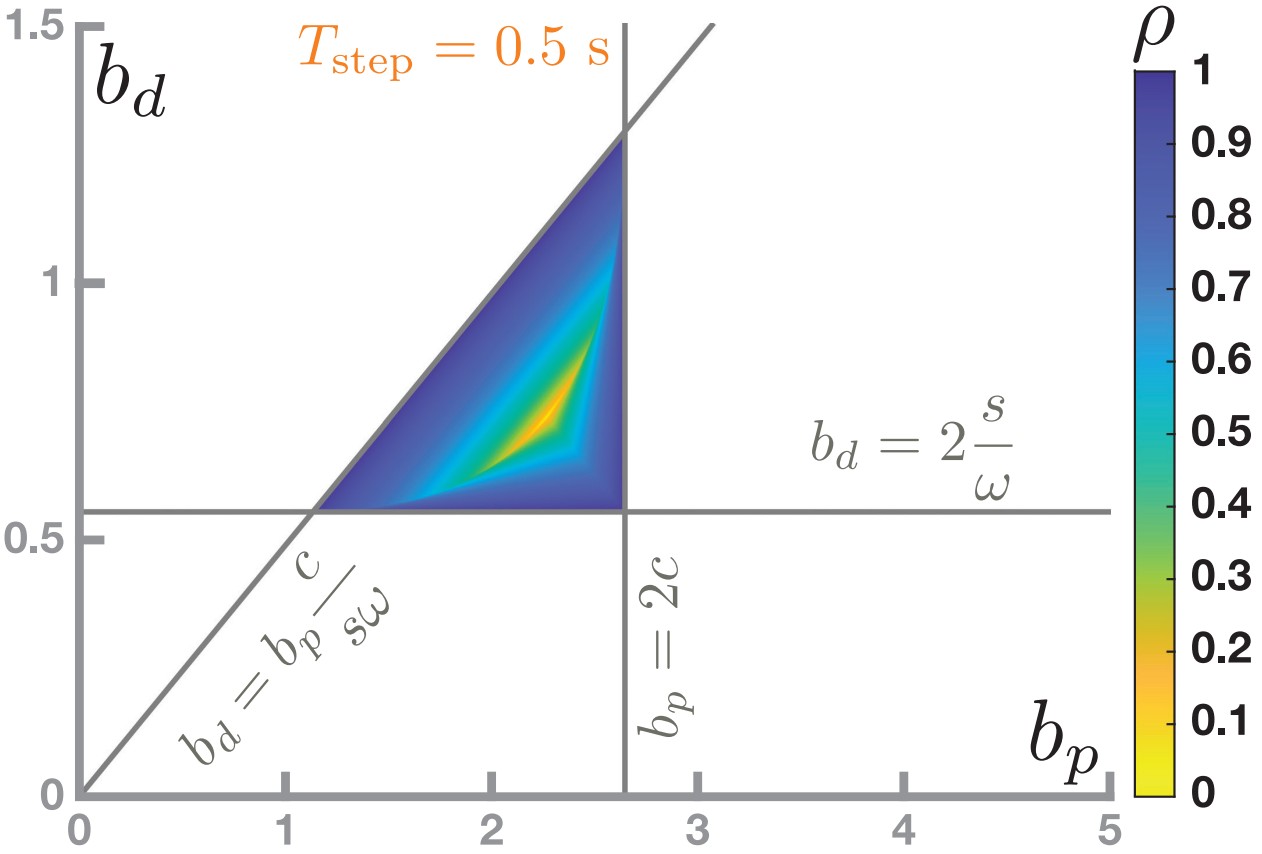

**Fig 5. Stability region in the parameter space spanned by $b_p$ and $b_p$.** White indicates that the system is unstable. The colored triangle is the region of stable walking systems, with color corresponding to the largest absolute eigenvalue, $\rho$, of the system's transition matrix $A$.

system can handle. In fact, it has repeatedly been shown that there is a very limited (if any) relationship between the local stability of a system, in terms of eigenvalues around a fixed point or limit cycle, and robustness, in terms of the size of a perturbation that can be handled without falling, for walking systems [20–22].

**Result 3** *Given $\omega$ and $T_{\text{step}}$, the region of stable control parameters is the set of control gains $(b_p, b_v)$ satisfying the following three inequalities*

$$2c - b_p > 0, \qquad 2\frac{s}{\omega} - b_d < 0 \qquad and \qquad b_d < b_p \frac{c}{\omega s} \tag{13}$$

Fig 5 illustrates this Result by showing how stability of the walking system over the parameter space spanned by $b_p$ and $b_d$, for a fixed step time $T_{\text{step}} = 0.5$ s. Color represents the largest absolute eigenvalue of $A$, which is referred to as the *spectral norm*, $\rho(A)$. White indicates instability ($\rho(A) \geq 1$). Other colors indicate stability, with brighter colors denoting smaller $\rho(A)$, implying that the system relaxes towards its periodic orbit faster after a perturbation. The pairs of control gains for which the walking system is stable form a triangular region in the parameter space spanned by $b_p$ and $b_d$, delimited by three lines. At each of these three lines, one of the inequalities in Eq 13 changes between true and false, and the inner triangle is the region in which all three inequalities hold.

## 3 Predictions and comparison to human data

In the previous section we have defined a simple walking system by combining the biomechanics of a single-link inverted pendulum with a simple point mass and a rigid leg with a proportional-derivative control law for the foot placement of steps that are taken after a constant time interval. Analysis of the system equations showed that for each parameter set, the walking system has one periodic orbit. Moreover, the analysis provided a criterion for the asymptotic stability of this periodic orbit. In this section, we will compare the implications of these theoretical results with actual human walking. We will make a prediction of how the neural feedback control gains should change between slow and fast walking and make preliminary comparisons with experimental data.

### 3.1 Stability changes with cadence

We showed in Result 3 above that for a given step time, the region of stable parameter combinations is a triangle in the space spanned by the two gains parameters of the proportional-derivative feedback controller for foot placement, $b_p$ and $b_d$. An example of this region for $T_{\text{step}} = 0.5$ s is shown in Fig 5. Fig 6A shows this triangular stability region in the parameter space spanned by $b_p$ and $b_d$ for two other step times of $T_{\text{step}} = 0.55, 0.75$ s, corresponding to stepping cadences of 110 and 80 steps per minute. These two examples indicate that the region of stable parameter sets shifts towards higher control gains for slower-paced walking. This means that when walking at a slower cadence, with longer step times, control gains need to be higher, or the system will not be stable. Similarly, when walking at faster stepping cadences, with shorter step times, control gains need to be lower.

Fig 6B and 6C shows how the stability region changes with cadence in more detail. Fig 6B plots the range of $b_p$-gains that are part of a stable parameters set changes against cadence. For

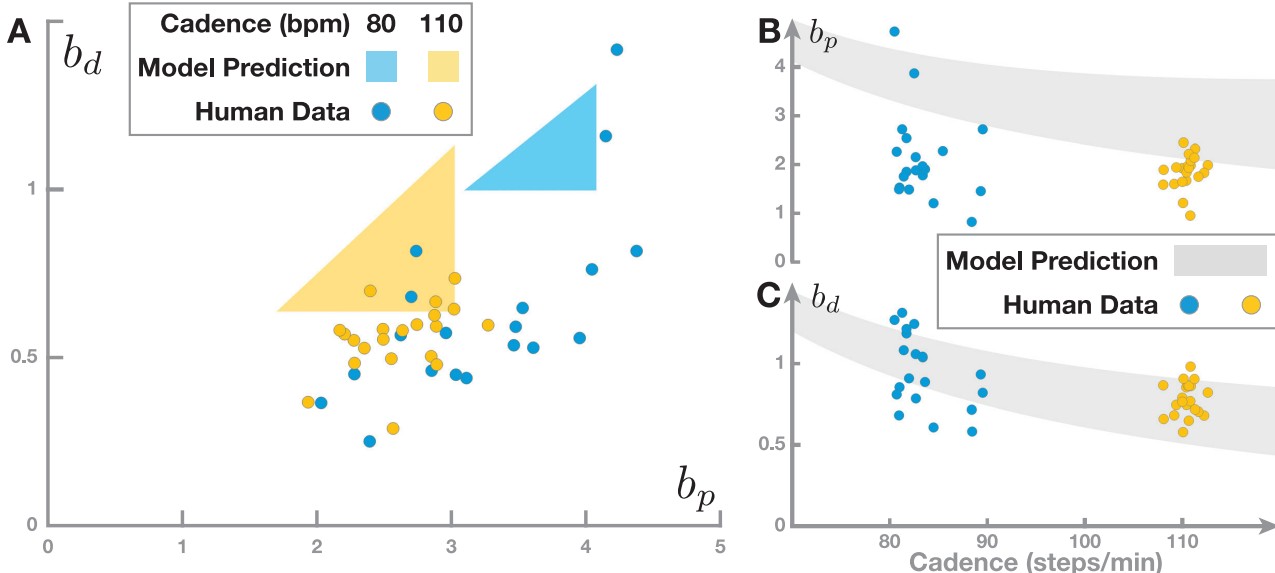

**Fig 6. The limits of the stability region change with cadence.** Panel A shows the triangular stability region for two example cadences, 80 (blue) and 110 (yellow) steps per minute. The dots show estimated parameter values from humans walking to metronomes at that cadence. The right side shows how the limits of the stability region change with cadence. The gray areas represent the projections of the stability region onto the $b_p$ (Panel B) and the $b_d$ axes (Panel C), vs. cadence on the horizontal. The dots are parameter estimates from the same human data as in Panel A, plotted here against the average cadence each participant walked at The cadence for each participant is the average for that trial, which usually differs by some degree from the paces imposed by the metronome.

each cadence value on the horizontal axis, all $b_p$-gains that are part of a stable parameter sets at this cadence are shown in gray on the vertical axis. This is essentially the bottom side of the triangular stability region at that cadence. As indicated by the two individual examples in Fig 6A, this graph shows that for decreasing cadence, the control gain, $b_p$, has to increase for the system to be stable. Fig 6C shows the same graph for the derivative control gain, $b_d$, vs. cadence. Similar as for the proportional gain, the projected stability region shifts to higher derivative gains, $b_d$, for lower cadences.

This theoretical result makes a prediction for human walking: *With decreasing walking cadence, control gains from CoM to foot placement should increase.* As can already be appreciated from the dots in Fig 6, this prediction seems to hold to some extent for human data. In the next section, we will discuss how to estimate control gains and compare such a prediction with experimental data.

## 3.2 Comparison with human data

The control law in our model determines the foot placement at each step based on sensory information about the CoM state at midstance (see Eqs 3 and 4). If we assume that humans use this control law, we can estimate the parameters from experimental data, as initially proposed and done by [13]. The approach, briefly, is to fit a linear regression model that predicts foot placement based on the CoM state at midstance. The slopes resulting from the linear regression for foot placement by CoM position, $\hat{b}_p$, and velocity, $\hat{b}_d$, are estimates of the control gains $b_p$ and $b_d$ (see Section 6 in the S1 Text).

To see how model predictions agree with actual human walking, we calculated the slopes, $\hat{b}_p$ and $\hat{b}_d$, for the medial-lateral direction from an existing data set from a previous experiment. Briefly, $N = 21$ neurotypical young participants walked on a self-paced treadmill to a metronome at two different cadences, 80 and 110 beats per minute, with resulting average walking speeds of 0.73 and 1.17 m s$^{-1}$. At randomized points throughout the trial participants received a perturbation in the form of a galvanic vestibular fall stimulus, but here we only analyze steps without such perturbations and after a wash-out period. For further details on the experimental protocol, we refer to [23].

Fig 6 shows the slopes estimated from this data set, relative to the stability region predicted by the model, for walking at 80 steps per minute (blue) and 110 steps per minute (yellow). Note that in the model simulations in Fig 6 we used a parameter value for the pendulum eigenfrequency $\omega = \sqrt{g\hat{l}^{-1}}$ based on the average leg length $\hat{l} = 0.77$ m from the 21 participants in the experimental sample, in order to make the model as similar as possible to the experimental data. The figure shows that the model stability regions predict the experimental estimates well in some aspects, but less well in others. For walking at 110 steps per minute, the estimated proportional gains, $b_p$, are well within the parameter range of the stability region, but the estimated derivative gains, $b_d$ are too low. For 80 steps per minute, the human parameter estimates are substantially more variable. The range of the position gains goes beyond the stability region on both sides, and the velocity gains are substantially lower than predicted by the model. Overall, this shows that the model predicts the position gains observed in humans reasonably well, especially for higher cadence walking. Moreover, the model showed the same trend for increasing cadence as found in humans. However, velocity gains were less well predicted by the model.

## 4 Discussion

We have presented a model of walking combining biomechanics and neural control of stability. Our general hypothesis was that a meaningful quantification of *stability* in walking requires

consideration of both these aspects, biomechanics as well as neural control, of walking. For biomechanics, we followed the well established linearized single-link inverted pendulum model (Eq 1) to approximate the body dynamics [8, 16, 17]. For neural control, we used a proportional-derivative controller for foot placement based on the center of mass state at mid-stance (Eqs 3 and 4), also following well-established ideas in the literature [13, 24]. With these equations, a walking system is parameterized by the proportional-derivative control gains, $b_p$ and $b_d$, the step time, $T_{step}$, the foot placement offset, $b_o$ and the body pendulum eigenfrequency, $\omega$. We analyzed a walking system by characterizing the periodic orbit of each system (Result 1), finding a condition for stability of this periodic orbit (Result 2) and determining the combinations of control parameters for which the system is stable (Result 3). Finally, we compared model predictions with experimental data from previous studies.

## What does *stability* mean?

The term *stability* refers to several different but overlapping concepts. Most generally, something is *stable* if it is unlikely to change in a substantial way. A fall is certainly a substantial change, so we call a period of standing or walking stable if it is less likely to include a fall. This characterization is indirect, because any particular period of walking or standing either does or does not include a fall. Since falls are rather rare events in practice, directly estimating the likelihood of a fall for a specific person and activity would require a large number of observations, which is not feasible in practice. A roundabout way of doing this is to estimate the fall risk for certain groups of people, like older adults or populations with specific neuromotor impairments, by comparing the incidence of falls over longer time periods with control groups. One problem with such estimates is that it is impossible to control what people do over a long period of time, and a difference in activity can easily lead to a difference in fall incidence. Nonetheless, such fall risk estimates provide meaningful differences between groups.

While group-level effects are informative, it is important to understand *why* certain groups are less stable and tend to fall more often than others. To that end, we study stability over shorter periods of time, where it is easier to control other factors, mostly by imposing certain conditions in a laboratory. Since falls are both rare and dangerous, we usually observe other effects that are also associated with stability. In standing, the *need to take a step* is a substantial change of behavior that is easy to observe and can be made to happen frequently by designing an experimental protocol around it. This is the basis of the *Margin of Stability* measure, which characterizes how close the biomechanical state of the system is to the threshold of having to take a step. In walking, taking a step is part of the normal behavior and *not* a substantial change from it. Nonetheless, the MoS has been frequently used to characterize stability in walking. This mismatch is mostly due to the lack of alternative measures, and it has been widely recognized [9] and has led to some "paradoxical" results [10].

In dynamical systems theory, *asymptotic stability* is a well-defined property of a fixed point or periodic orbit of a system [25]. A "substantial change" is defined here as *any change at all following an infinitesimally small perturbation after an infinitely long time*. This definition, and associated methods from dynamical systems theory, have been widely applied to the analysis of walking movements, mostly in the form of measures that quantify how fast a system relaxes back towards the stable state after a perturbation, e.g. Lyapunov exponents [26], Floquet multipliers [27].

In our analysis of walking as a dynamical system with biomechanical and control components presented here, we have adopted the dynamical systems definition of stability. We use a definition of stability which is closely linked to Floquet multipliers, to ask whether a periodic orbit is stable or not, depending on the parameters. This question has a different quality than

the analysis of experimental data from human walking. For actual walking, the question *whether* it is stable is an extreme case, because the actual loss of stability is a rare and extreme event. A more reasonable question to ask is *how stable* a specific period of walking is, i.e. to measure a degree of stability. For a walking system as modeled here, in contrast, we have to ask *whether* it is stable before we can meaningfully quantify a degree of stability. We stress that in the model analysis presented here only answered the first question. We did not, in any way, provide a measure for a *degree of stability*. Specifically, our work does not offer any measure that is applicable to experimental data in the way the MoS is. While the MoS has several drawbacks, as discussed in the introduction, it is currently the most useful way we have to quantify stability of actual walking humans.

## How is the model useful?

If the model analysis here does *not* measure a degree of stability, how is it relevant to our understanding of human walking? In the spirit of "all models are wrong, but some are useful" [28, 29], we contend that our walking model is both wrong and useful. One way in which a model can be useful is to make predictions. As shown above, our model predicts that control gains should change with walking cadence (see Fig 6). Specifically, when walking at slower cadences, the control gains have to be higher. This prediction is at odds with a result from [13], which found no significant difference between regression slopes at three different speeds (see Section 6 in the S1 Text for the relation between control gains and regression slopes), but in line with our own data (presented here in Fig 6). Humans generally vary their cadence in direct proportion to their step length, and modulate both together to vary walking speed [30], so our model predicts an effect of walking speed that [13] did not observe. One thing to point out here is that [13] showed the *absence of significance*, rather than the *significance of absence* of the effect of walking speed. The speeds in their protocol were 1.0, 1.2 and 1.4 m s$^{-1}$, which is not a large range, and all of these speeds can be reasonably described as "medium". Moreover, the protocol controlled speed, and different people will use different cadences to walk at the same speed. As mentioned, our analysis of data from a previous experiment, in which cadence was controlled, supports that decreased cadence leads to increased control gains, as we found slopes for position being higher at 80 vs. 110 beats per minute (Fig 6, paired t-test, $p < 0.001$), although there was no significant difference in the velocity slopes ($p = 0.39$). However, this study also was not specifically designed to test this hypothesis, and included only two cadences. A study by [31] analyzed walking on a treadmill at a wider range of speeds by fitting regression models, and results show that the $R^2$ values, i.e. the amount of foot placement variance explained by the CoM state at midstance, changed with speed, but this study does not report the regressions slopes, which estimate the control gains. Based on the evidence, we conclude that our model makes a prediction that is partially in agreement with current understanding and that more research is needed to test whether foot placement control indeed changes with walking speed.

Our model also predicts specific values for the feedback gain parameter. Comparison with experimental data in Fig 6 shows that the model predictions do not match the observations in some aspects. While the parameter estimates for the proportional gain, $\hat{b}_p$, are within or reasonably close to the predicted interval, the derivative gain estimates, $\hat{b}_d$, are substantially lower than the values predicted by the model. Our modelling results and the experimental data clearly do not fully match, and it may be interesting to consider why this is so. This mismatch could either be due to the experimental design introducing unintended effects, e.g. from walking on a treadmill, or from the model missing some key aspects of human walking. Below, we consider three potential reasons for why the model might be wrong in this aspect. In short,

they are (1) the assumption that collisions do not matter, (2) the assumption that the pendulum eigenfrequency, $\omega$, is constant, whereas effective leg lengths varies between participants, and (3) the model does not include ankle roll control.

**Energy loss.**   The first reason why our model may show results which are not in agreement with human walking is that our model does not contain collision dynamics [32]. It is well known that at heelstrike, energy is lost from the collision between the foot and the ground [33]. Such a loss of energy would shift the actual place where the foot needs to be placed to come to a standstill backwards (inwards in the frontal plane). However, in humans, during steady state walking, such a collision loss is compensated for by an equal injection of energy due to push off. While it is possible to write out the equations while accounting for such collisions, doing so would yield equations which can only be solved numerically [32]. Moreover, energy loss will be compensated for over time on average, as stated above, so we expect that neglecting collision dynamics has limited effects on our results.

**Effective leg length.**   The second reason why our model results show limited agreement with the human results is that we assumed the eigenfrequency of the body as an inverted pendulum, $\omega$, to be constant. While this is a common assumption, the effective leg length, i.e. the distance between the body CoM and the contact point, changes substantially throughout the gait cycle. The common understanding is that the effective leg length is modulated to minimize vertical CoM motion [30], but modulation of the CoM height can also be used to affect the CoM dynamics, making it a candidate for a feedback control mechanism [34]. We are currently unaware of any experimental evidence that humans might actually use leg length modulation as a control mechanism during walking, in contrast to ankle roll (see below). In our comparison of model predictions with experimental data, we used the average effective leg length of the human participants, $\hat{l} = 0.77$ m. The eigenfrequency parameter, $\omega = \sqrt{g\hat{l}^{-1}}$, affects the stability of the system. Specifically, with increasing leg length the triangular stability region grows larger, and also shifts downward and to the left. This effect is relatively small, however, indicating that even if humans use active modulation of effective leg length as a control mechanism, the gain parameters for foot placement control would be largely unaffected by that. Here we note that not accounting for this relationship by using a single group estimate for $\omega$ might affect how the model prediction compares to the experimental data, but a detailed investigation of this effect would go beyond the scope of this manuscript.

**Ankle roll.**   The third aspect potentially responsible for the observed discrepancy between model predictions and experimental data is ankle roll control, which, we will argue, is likely to be the main factor. Ankle roll is a control mechanism for stability during walking, where humans actively use lateral ankle musculature during single stance to move the body in a desired direction [35–40]. Ankle roll and foot placement control interact with each other. A model study showed that adding ankle roll control resulted in reduced use of the foot placement mechanism in response to a simulated perturbation [41]. Human experiments showed similarly that larger lateral ankle responses to a perturbation are correlated with smaller foot placement changes [42]. The relation between ankle roll and foot placement also changes with cadence. Humans showed larger foot placement responses to a perturbation at high vs. low cadence, and larger ankle roll responses at low vs. high cadence [23]. To sum up, these results indicate that ankle roll is an important aspect of human stability control, and that it is more relevant at slower walking (although, see [39]). The model presented here does not have ankle roll control. We postulate that due to this lack of ankle roll, the walking system is overall less stable and requires higher foot placement control gains than those observed in humans. In other words, humans need less foot placement control because they already stabilize with ankle roll to some degree, and this difference is more prevalent in the low-cadence condition.

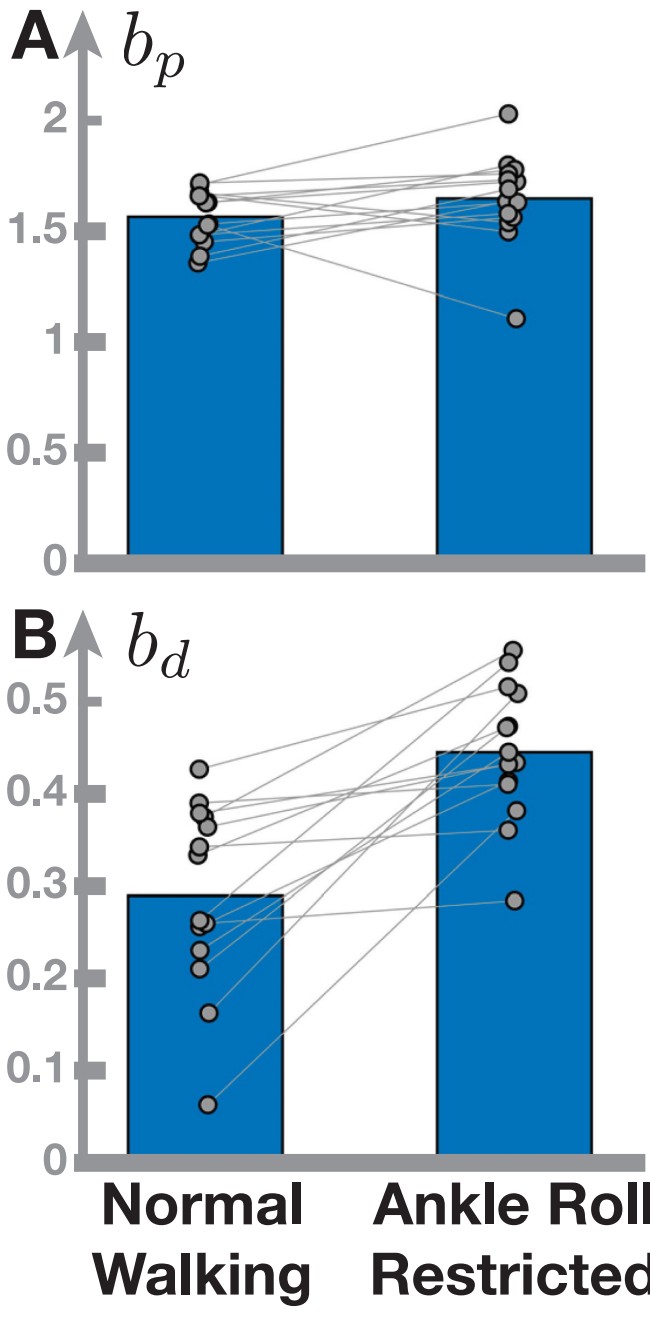

**Fig 7. Gain parameter values estimated from human data change when restricting the ankle roll mechanism.** Bars show the average proportional (Panel A) and derivative (Panel B) gain parameter estimate from N = 30 participants, with normal walking on the left and ankle roll restricted by a ridge attached to the sole of the shoes on the right. Gray dots represent individual participants, with gray lines connecting data from the same participant in both conditions.

To explore the hypothesis that ankle roll allows smaller foot placement gains, we re-analyzed data from an experiment where ankle roll was restricted. $N = 30$ healthy participants walked on a treadmill at normal and slow speeds, wearing either standard shoes or shoes with a ridge below the sole that restricted the contact surface of the foot and effectively removed the

effect of ankle roll moments on stability. For further details on the experimental protocol, please refer to [43]. Fig 7 shows data from walking at normal speed ($1.25\sqrt{l}$ m s$^{-1}$), with and without the ankle roll restriction. The position slopes are similar between the two conditions (t-test, $p = 0.16$). The velocity slopes are substantially higher in the condition with restricted ankle roll ($p < 0.001$). This indicates that with access to ankle roll control, humans use lower derivative gains for foot placement control, supporting our hypothesis that the lack of ankle roll in the model is the main reason for the discrepancy between model prediction and experimental data.

## Limitations

There are different ways in which the model is wrong that limit what we can infer from comparisons between model predictions and experimental data. When analyzing the relationship between the body CoM state and foot placement, the CoM is often approximated by the position of markers on the posterior-superior iliac spine [44]. Although this is a reasonable approximation, there is still considerable difference between these pelvis markers and the whole-body CoM. This error between the approximation from pelvis markers and the true CoM might affect the slope estimate. Midstance is usually defined functionally, as the pelvis crossing the stance foot ankle. In our model, midstance is defined as the temporal mid-point between two heel-strikes. This difference might introduce a bias in the regression slope estimates of the control gains. Midstance was initially chosen as a point of interest because this is where the explanatory power of the CoM state predicting the foot placement location reaches a plateau [13]. But a later study by [31] shows that this is not necessarily the case for slow walking, where the $R^2$ values keep rising substantially during the later part of the step. Estimating the state of the body CoM, and constructing a descending motor command and physically moving the swing leg according to a neural control law takes time. In reaching, goal-directed updates of a movement during execution can be as fast as 50–100 ms after a stimulus [45], while voluntary movements are generally understood to have a delay of >100 ms [46]. For normal and fast walking, the time between midstance and heelstrike is not much longer, so it is reasonable that the explanatory power of the CoM for foot placement would plateau around midstance. For slow walking, however, the time between midstance and heelstrike is substantially longer, and there is no need to cease control arbitrarily at midstance. A later time point might be more suitable to calculate regression slopes between CoM state and foot placement estimate control gains in slow walking, but how such a point in time should be determined is an open question. In fact, if humans actually estimate CoM state in some way, and how they do so, are still open research questions. While it is well known that visual [37, 47], vestibular [48–51], and proprioceptive systems [52] all play a role in the control of foot placement (see also [18] for a review), it is unknown if, and how, these are integrated into an estimate of center of mass state. Nonetheless, in modeling work, center of mass state seems to predict balance responses better than joint level feedback, suggesting that sensory sources are integrated in some way [53].

The fact that high $R^2$ (up to 0.95) are consistently reported when estimating velocity and position gains from human data [13] suggest that these parameters can be estimated with good precision from the data. Here we found adjusted $R^2$ values around $0.754 \pm 0.0966$ (mean ± standard deviation) for normal walking to a metronome at 110 beats per minute, and $0.666 \pm 0.132$ for slow walking at 80 beats per minute. Between the two predictors, the CoM position at midstance tends to be have more explanatory power for the foot placement than the velocity. When using either variable as a single predictor, the adjusted $R^2$ values for normal walking drop to $0.497 \pm 0.158$ for the model with position as single predictor and $0.246 \pm 0.129$ for velocity. However, the velocity still has substantial explanatory power and is significant as a

predictor ($p < 0.0001$ for all subjects). This indicates that even though $b_d$ is not well predicted by the model, and the estimates tend to vary more between subjects, the CoM velocity is important for the neural controller to adjust foot placement location with the goal of maintaining upright stability.

Our theoretical approach analyzed stability by determining parameter values for which the periodic orbit of the walking system is asymptotically stable. It is tempting to infer varying *degrees* of stability from this analysis. Fig 6 shows that the triangular stability region of the parameter plane spanned by the two proportional-derivative control gains, $b_p$ and $b_d$, is smaller for slow-paced walking. This does *not* imply that slow-paced walking is less stable. Comparing stability in different systems would require definition of a stability measure. One option for such a measure would be the spectral radius of the transition matrix, $\rho(A)$, which is already used to determine the binary notion of $\rho < 1$ indicating *stable* and $\rho > 1$ indicating *unstable* we used here. Could we extend this notion of stability to one where smaller spectral radius indicates a more stable system? This is possible in principle, but it is questionable how useful such a definition would be in practice (see for instance [20, 21]). As shown above, there are substantial discrepancies between the predicted stability region and estimated gain parameter values. This implies that if we fitted model parameters to experimental data in order to measure stability with the spectral radius of the fitted model, this model would not actually be stable, and it would be unclear how the spectral radius measure relates to stability of the walking human. One could argue that a larger stability region is preferable, as it would allow for a more flexible system, in which larger fluctuations can be handled. If motor or sensory noise are taken into account, we could even speculate that their effects would be similar to changes in control gains at each step, and hence systems which have larger stable regions for control gains are more likely to be robust against noise. Taking these speculations one step further, one could argue that having control gains which are at the borders of the stability region is unfavorable, as changes in the control gains would be more likely to result in leaving the stability region. However, as of yet, this is speculation, and testing these ideas would require the addition of motor and sensory noise to these models. Moreover, we have seen that the estimated control gains of humans typically do not fall within the regions specified by the model, and we would need large sets of human data to see the typical stability regions in humans. Potentially, those human participants that have control gains which are at the borders of the healthy human population are then at a higher risk for falls. There is some evidence that indeed such differences in control gains are related to pathologies [54]. For these reasons, while a stability measure based on this model, or a similar one, could be very useful, such a model would have to be fully validated and explain the human data more closely than the version presented here as a starting point.

We assumed that the step time, $T_{\text{step}}$, is chosen as a parameter and then kept constant, rather than modulated from step to step. However, modulating the step time based on the CoM state could also be used as a feedback mechanism. Larger step time will result in faster movement of the CoM at the point when a step is eventually taken, because the CoM had longer time to fall and build up speed. It is conceivable to use this effect as a feedback mechanism for stability control, where the step time is modulated based on the CoM position and velocity at midstance. Neither step time, nor modulation of $\omega$ via effective leg length or ankle roll, as described above, are included in the current model, despite ankle roll being well supported as a control mechanism in the literature. Further research is required to investigate how the addition of these mechanisms would affect stability, and whether humans might actually modulate their CoM height and step timing as predicted by these mechanisms.

Humans live in a three-dimensional world and walk on two-dimensional surfaces. The approximation of the body as a single-link inverted pendulum allows to neglect the vertical

dimension and analyze the two horizontal dimensions separately. A small difference in the control law of alternating the offset term, $b_o$, changes from describing progressive walking in the anterior-posterior direction to alternating walking in the medial-lateral direction (Eqs 3 and 4). Interestingly, the subsequent analysis shows that the stability criteria are identical for both of these two variants of the walking system (Result 2), implying that stability is the same in both directions. This is in direct contrast to common understanding in the field that maintaining stability in the frontal plane is more challenging than in the sagittal plane. Kuo analyzed the passive walking in two planes and found that while motion in the sagittal plane can be passively stable when walking on a ramp with a specific incline, frontal plane motion is unstable [55]. Kuo and Bauby study directional stability in humans [56], finding that foot placement variability is larger in the frontal vs. the sagittal plane, and frontal plane variability increases more when closing the eyes. Furthermore, visual perturbations affect foot placement variability in the frontal plane substantially more than in the sagittal plane [57]. Our model fails to predict this difference between frontal and sagittal plane stability, because the criterion for the stability of a walking system, stated in Result 2, is the same for the progressive and alternating version of the walking system. This implies that the reasons for the difference between frontal and sagittal plane stability reported by Kuo et al. are not based on alternating vs. progressive stepping, but rather due to features not currently included in our model, such as the foot being much longer than it is wide, or the neural control parameters being different for the two directions.

As discussed above, ankle torques play a substantial role in balance control, and ankle torques in the sagittal plane have a much higher range due to the increased moment arm along the foot. The system analyzed by Kuo and colleagues [55–57], following [33], contains legs, hip joints and roll motion, leading to more complex dynamics compared to the linearized single-link inverted pendulum analyzed here. These passive dynamics contribute to the stability of the system via interaction forces and reduce the need for active control, to the point where, in certain specific configurations, the sagittal plane is passively stable. Adding these factors to the model analyzed here is a subject of future work.

## Relationship to other modeling studies

Many studies have analyzed walking using simple inverted pendulum models, both in human motor control and in robotics. Hof introduced the concept of the extrapolated center of mass [8] and suggested possible controllers to walking that maintain balance by modulating either the step time or the foot placement location [12]. For the latter, Hof proposed placing the foot at a constant offset from the extrapolated center of mass. This controller can be expressed as a special case of the control law analyzed here (Eqs 3 and 4), with control gains $b_p = c + s$ and $b_d = \omega^{-1}(c + s)$, where some transformations are required due to Hof using the state at foot placement, rather than at midstance, as input. These parameters are well within the stability region (see Fig 5), with spectral norm $\rho = 0.21$ for the example used here. Roboticists have thoroughly studied similar simple models. For instance, Koolen et al. analyze stability and "capturability" of a mechanically similar model in general terms, determining regions of state space from where the robot can be brought to a stand within a certain number of steps [58]. Similarly, Zaytsev et al. analyze "viability", i.e. the ability to avoid falls or, more generally, failures, for both the compass walker with push-off and the linearized inverted pendulum [19]. This line of work is different from ours in that it is mainly mechanical and agnostic to the controller, asking whether a specific state of the system is capturable or viable assuming the best possible control actions. Thus, it provides relevant boundary conditions and complementary information to our work. Modeling studies in human motor control, on the other hand, are often analyzing

specific controllers. Joshi and Srinivasan designed a controller for foot placement and push-off for a compass walker and fitted the parameters to human data from experiments with sizable mechanical perturbations [59]. Their parameter estimates for foot placement, accounting for dimensionality, are within the stability region of our model, though the fact that cadence was not controlled precludes a detailed evaluation. Patil et al. analyzed a compass walker with different task-level controllers for push-off and found that the basin of attraction is largest and most regular using a closed-loop controller for speed [60]. Furthermore, a relatively low number of controllers with different target speeds covers the viability kernel, i.e. the region of state space where the system can continue walking indefinitely [61], and they suggest that humans might maintain balance by switching between pre-learned controllers using high-level, executive control. This work shares our main goal of determining how humans control walking. A critical difference is that Patil et al. use an optimal control approach to consider different task-level controllers for speed and position in the anterior-posterior direction, and analyze the stability as an incidental property of such a controller, while we use a linear proportional-derivative controller with the main goal of maintaining stability. The major novel result of this study in comparison to these other works is that we show that when using a PD controlled linear inverted pendulum, there exists a limited region in parameter space where walking is stable, and that this region shifts as a a function of cadence.

## Supporting information

**S1 Text. Provides intermediate results and detailed derivations of the main results.**
(PDF)

## Author Contributions

**Conceptualization:** Hendrik Reimann.

**Data curation:** Hendrik Reimann.

**Formal analysis:** Hendrik Reimann.

**Investigation:** Hendrik Reimann, Sjoerd M. Bruijn.

**Methodology:** Hendrik Reimann.

**Software:** Hendrik Reimann.

**Validation:** Hendrik Reimann.

**Visualization:** Hendrik Reimann, Sjoerd M. Bruijn.

**Writing – original draft:** Hendrik Reimann.

**Writing – review & editing:** Hendrik Reimann, Sjoerd M. Bruijn.

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
