## [Decision Letter · Decision Letter 0]

31 Jul 2023

Dear Dr. Reimann,

Thank you very much for submitting your manuscript "The condition for dynamic stability in humans walking with feedback control" for consideration at PLOS Computational Biology.

As with all papers reviewed by the journal, your manuscript was reviewed by members of the editorial board and by several independent reviewers. The reviewers were generally enthusiastic about the approach of combining biomechanics and neural control within a single model to analyze the stability of walking. However, the reviewers expressed a range of concerns that should be addressed. In light of the reviews (below this email), we would like to invite the resubmission of a significantly-revised version that takes into account the reviewers' comments.

We cannot make any decision about publication until we have seen the revised manuscript and your response to the reviewers' comments. Your revised manuscript is also likely to be sent to reviewers for further evaluation.

Sincerely,

Adrian M Haith

Academic Editor

PLOS Computational Biology

Thomas Serre

Section Editor

PLOS Computational Biology

Reviewer's Responses to Questions

**Comments to the Authors:**

Reviewer #1: This manuscript investigates an inverted pendulum model of walking with neural feedback control in the form of the stepping strategy. The concept is quite simple (and elegant) by combining two established methods – simple inverted pendulum models and foot placement control based on mid-stance. The description and development of the model were thorough and easy to follow. Although some theorems were unsurprising for those familiar with the models, the authors gave clear explanations for those that might be less familiar. The control gains and cadence finding is also very interesting and relevant to open questions in the field as well. The comparisons with human data as presented, however, left me unconvinced about the validity of the model in comparison to human behavior. Yet, the authors did an extensive job of outlining the utility and deficiencies of the model which was much appreciated.

1. Experimental data comparisons: Previous data was used to evaluate (very qualitatively) whether the model could predict human behavior. While I believe the human comparisons are probably necessary, they currently weaken the paper in my view. In Figure 5A, there is considerable scatter in the lower cadence data. While the shift in bd was later explained, the wider scatter in bp was not. In addition, comparing the model against human data at more than two cadences would be beneficial. A quantitative method of comparisons would also lend more strength to the arguments. I do not doubt the validity and usefulness of the model, but barring gathering more data, I would almost be more interested more broadly in the model’s predictions, not just with respect to cadence.

2. One notable feature missing from this model is collisional losses. As also noted in Lines 311-312, the model has no method of dissipating energy which can build up. A nice explanation was provided in the Discussion, but the argument for why the effect of collision is small is unconvincing. Although push-off can make up for this energy loss, collisional losses have a substantial effect on system dynamics. Impacts might also be another reason for the difference between the current model and the Bauby and Kuo model, rather than model complexity.

3. The sections on energy loss, effective leg length, and ankle roll were informative in outlining why model predictions might be wrong. It was also interesting that low bd could be explained by the lack of ankle roll. However, the effect of the other two missing items seemed to be treated more dismissively. On a similar note to the prior concern and without going outside the scope of the paper, could the authors give some indication whether a constant leg length is expected to have a large or small effect on model predictions? Otherwise, it currently reads that the ankle roll was already decided to have the largest effect based on the author’ past work, which might be true, but minimizing the other two items still requires more support to be convincing to readers.

4. Figure 5 – the caption notes that the human data in Figure 5BC is the same as from Figure 5A. Both are from a previous experiment where individuals walked at a set cadence dictated by a metronome. If so, why are there variations in the human cadences for Figure BC?

5. What was the approximate speed range at these two cadences? Were they comparable between human data and model? And among the studies discussed in "How is the model useful"?

6. Line 503 states that humans vary cadence "in direct proportion" to speed. However, there is also an established power law that links speed and step length (and cadence) by [1] (see also [2]), resulting in a more curvilinear relationship.

Minor comments:

If cadence is to be the primary focus, it should be noted, perhaps in the last paragraph of Introduction.

Line 190 – how were the initial conditions chosen "exactly right"?

Line 572 – Typo ")although"

[1] D. W. Grieve, "Gait patterns and the speed of walking," Biomed. Eng., vol. 3, pp. 119–122, 1968.

[2] S. H. Collins and A. D. Kuo, "Two independent contributions to step variability during over-ground human walking," PLoS ONE, vol. 8, no. 8, p. e73597, 2013.

Reviewer #2: Manuscript Number: PCOMPBIOL-D-23-00629

The condition for dynamic stability in humans walking with feedback control

SUMMARY OF WORK:

This manuscript addresses the very important topic of stability during human walking developing a simple model that incorporates both (bio)mechanics and neural control and then employing it to make predictions about the contribution of center of mass state feedback to dynamic balance at slow vs faster walking speeds. These predictions are tested using data from relevant human walking experiments in a laboratory setting.

First the authors clearly present mathematical descriptions of both the mechanical system (i.e., a two-legged, 2DOF (fore-aft + medio-lateral) passive inverted pendulum (Eqs. 1, 2)) and control system (i.e., a proportional-derivative feedback control law that maps center of mass (CoM) state at midstance to an updated contact point of the foot on the ground relative to the CoM (Eqs. 3,4) and show output of walking simulations driven by the model (Fig. 1).

Next, the authors mathematically define periodic orbits and their stability to changes in control gains (Figs.2, 3) culminating in the key result describing the range of feedback control parameters that yield stable walking for a fixed step time (Fig. 4).

Then, the authors test their model’s prediction that slower walking (i.e., longer step time) should require higher feedback control gains against model-fits from human walking experiments at different walking speeds. The simple model does a reasonable job at predicting what humans do (Fig. 5).

Finally, the authors provide a comprehensive critique of their own model by addressing many, if not all, of the key limitations and how they might influence predictions about stability during real human walking (e.g., contribution of ankle roll to modulate CoP (Fig. 6). In doing so, the authors demonstrate how their model is ‘wrong, and useful’- a welcome approach that is all too commonly lost in the literature in favor of presenting more complicated models with ‘cleaner’ results, often at the expense of deeper understanding of the system.

In short, the manuscript presents a ‘first of its kind’ simple neuro+mechanical model that, as the authors would say themselves, is ‘wrong, but incredibly useful’. Their formulation acknowledges, and creatively merges previous mathematical descriptions of the mechanics and (neural) control of walking, with a keen focus on clearly and rigorously defining a set of formal conditions for dynamic stability. A strength is that the model is used to make testable predictions that are directly addressed with relevant human experiments. The interpretation of their results with respect to model limitations is especially strong. In this reviewer’s opinion, this work is truly exciting and stands to have great impact as a ‘go to’ foundation for how scientists, clinicians and engineers define, understand and work to preserve or augment balance in two-legged locomotion systems – not just in humans, but also robots, and humans wearing robots. The manuscript already offers a fine contribution as is, but below I offer a handful of comments that may help to further strengthen the clarity of the presentation and expand the scope of the discussion – two aspects that could increase the impact of the work.

COMMENTS:

1. As stated above, the modeling approach to merge the mechanics and control of walking and application of formal analysis to define stability are both excellent and clearly described in text and using mathematical formulas. What is missing, however, is a clear schematic description of the elements of the model at the front-end of the main text. One potential version of this would show a ‘cartoon’ that defines the inverted pendulum geometry and states and a block diagram that represents the feedback control. Introducing the phase space representation in this schematic with and example of stable and unstable orbits would also help frame the reader’s expectation early.

2. Along the lines of point 1. above, it would be helpful to have a clearer understanding of what the parameter b0 is? Is this akin to a baseline relative pose of the legs on the ground? i.e. a baseline ‘base of support’? A pictorial representation of the system would help clarify this.

3. Linked to point 2. above, in some cases in the presentation of results it is unclear whether both A/P and M/L directions are being addressed (e.g., sensitivity to control parameters b0, bp, bd – Figs. 3, 4, 5). It is suggested to tighten and clarify throughout about whether/when AP vs. ML are being addressed. Perhaps using different subscripts or subplots?

4. Does it matter how the perturbation is applied? For example, is perturbing via a change in initial conditions (Fig. 1) the same as a perturbation from a fixed periodic orbit (Fig. 3)? Is a perturb in position the same as a perturb in velocity? Does the phase and direction of the perturbation matter? If so, how would these issues affect the stability analysis throughout?

5. In reading the manuscript, this reviewer noted many model simplifications that the authors would do well to address (e.g., massless limbs, no joints, no swing leg dynamics, no arms, no angular momentum!; -- no explicit force production of the legs on the ground, no double support period, no collision losses; -- a hyperfocus on midstance state as the discrete control input). But, by the end of the paper, the authors had handled most if not all of these – often pointing to model modifications that could add complexity if/when needed.

One aspect that could use a bit more discussion is how the feedback control law presented in this manuscript would work in the physiological system. What sensory modalities are involved, how are they integrated to give CoM position and velocity? What about integral control? (i.e. the ‘I’ in PID).

6. The authors mention the need for ‘large sets of human data’ to ‘see the typical stability regions in humans’. There is a recent example of this, an open-source data set created by Jennifer Leestma at GaTech that may be useful to cite or even use in future collaborative work.

Biomechanics of locomotion during ground translation perturbations

JK Leestma, PR Golyski, CR Smith, GS Sawicki, A Young

Georgia Institute of Technology

Linking whole-body angular momentum and step placement during perturbed human walking

JK Leestma, PR Golyski, CR Smith, GS Sawicki, AJ Young

Journal of Experimental Biology 226 (6), jeb244760

Reviewer #3: The review is uploaded as an attachment.

**Have the authors made all data and (if applicable) computational code underlying the findings in their manuscript fully available?**

Reviewer #1: **No: **Yes, for the code. However, not for the human data. Cannot access "foot_placement_ml_by_com_50_CAD.csv" for instance when running the Matlab code.

Reviewer #2: **No: **The code that runs the walking simulations could be shared.

Reviewer #3: Yes

PLOS authors have the option to publish the peer review history of their article (what does this mean?). If published, this will include your full peer review and any attached files.

Reviewer #1: No

Reviewer #2: **Yes: **Gregory S. Sawicki

Reviewer #3: No
---

## [Decision Letter · Decision Letter 1]

19 Oct 2023

Dear Dr. Reimann,

Thank you very much for submitting your manuscript "The condition for dynamic stability in humans walking with feedback control" for consideration at PLOS Computational Biology. As with all papers reviewed by the journal, your manuscript was reviewed by members of the editorial board and by several independent reviewers. 

The reviewers overall were very satisfied with the responses and revisions, and had only relatively minor comments. Reviewer 3 in particular had a number of remaining issues that require clarification. We would therefore like to invite you to submit a further minor revision which addresses these remaining comments through revisions to the paper and/or responses to the reviewers' comments.

Sincerely,

Adrian M Haith

Academic Editor

PLOS Computational Biology

Thomas Serre

Section Editor

PLOS Computational Biology

Reviewer's Responses to Questions

**Comments to the Authors:**

Reviewer #1: The authors have addressed my concerns in their thorough responses. I have a few minor suggestions.

Figure 1 is a very helpful overview, but a more detailed caption might also help the reader to define some of the variables or know broadly what the components mean without searching in the text.

I do not think my comment about data accessibility was relayed to the authors, so I have reiterated it here. The human data is currently inaccessible, e.g., cannot access "foot_placement_ml_by_com_50_CAD.csv" when running the Matlab code.

Reviewer #2: Great job addressing feedback in the revised submission. Looking forward to citing this when it is published.

Nice work.

Reviewer #3: Uploaded as attachment.

**Have the authors made all data and (if applicable) computational code underlying the findings in their manuscript fully available?**

Reviewer #1: **No: **Yes, for the code. However, not for the human data. Cannot access “foot_placement_ml_by_com_50_CAD.csv” for instance when running the Matlab code.

Reviewer #2: Yes

Reviewer #3: Yes

PLOS authors have the option to publish the peer review history of their article (what does this mean?). If published, this will include your full peer review and any attached files.

Reviewer #1: No

Reviewer #2: **Yes: **Gregory S. Sawicki

Reviewer #3: No

Figure Files:

Data Requirements:

Reproducibility:

References:

---

## [Decision Letter · Decision Letter 2]

24 Jan 2024

Dear Dr. Reimann,

We are pleased to inform you that your manuscript 'The condition for dynamic stability in humans walking with feedback control' has been provisionally accepted for publication in PLOS Computational Biology.

Best regards,

Adrian M Haith

Academic Editor

PLOS Computational Biology

Thomas Serre

Section Editor

PLOS Computational Biology

Reviewer's Responses to Questions

**Comments to the Authors:**

Reviewer #3: The authors have adequately addressed my comments and I now believe the paper is suitable for publication.

**Have the authors made all data and (if applicable) computational code underlying the findings in their manuscript fully available?**

Reviewer #3: Yes

PLOS authors have the option to publish the peer review history of their article (what does this mean?). If published, this will include your full peer review and any attached files.

Reviewer #3: No

---

## [Editor Report · Acceptance letter]

22 Feb 2024

PCOMPBIOL-D-23-00629R2 

The condition for dynamic stability in humans walking with feedback control

Dear Dr Reimann,

I am pleased to inform you that your manuscript has been formally accepted for publication in PLOS Computational Biology. Your manuscript is now with our production department and you will be notified of the publication date in due course.

With kind regards,

Zsofi Zombor
